ecology, genomics

phenology, population genomics, isolation by time, resource wave, Pacific herring

**Author for correspondence:**
Eleni L. Petrou
e-mail: elpetrou@uw.edu

# Functional genetic diversity in an exploited marine species and its relevance to fisheries management

Eleni L. Petrou[1], Angela P. Fuentes-Pardo[2], Luke A. Rogers[3], Melissa Orobko[4], Carolyn Tarpey[1], Isadora Jiménez-Hidalgo[1], Madonna L. Moss[5], Dongya Yang[6], Tony J. Pitcher[7], Todd Sandell[8], Dayv Lowry[9], Daniel E. Ruzzante[10] and Lorenz Hauser[1]

[1]School of Aquatic and Fishery Sciences, University of Washington, 1122 NE Boat Street, Seattle WA 98105, USA
[2]Department of Medical Biochemistry and Microbiology, Uppsala University, Uppsala, Sweden
[3]Fisheries and Oceans Canada, 8888 University Drive, Burnaby, British Columbia, Canada V5A 1S6
[4]Earth to Ocean Research Group, Department of Biological Sciences, Simon Fraser University, 8888 University Drive, Burnaby, British Columbia, Canada V5A 1S6
[5]Department of Anthropology, University of Oregon, Eugene, OR 97403, USA
[6]Department of Archaeology, Simon Fraser University, Education Building 9635, 8888 University Drive, Burnaby, British Columbia, Canada V5A 1S6
[7]University of British Columbia, Institute for the Oceans and Fisheries, Vancouver, British Columbia, Canada
[8]Washington Department of Fish and Wildlife, 16018 Mill Creek Boulevard, Mill Creek, WA 98012-1541, USA
[9]Washington Department of Fish and Wildlife, 1111 Washington Street SE, 6th Floor, Olympia, WA 98504-3150, USA
[10]Department of Biology, Dalhousie University, Halifax, Nova Scotia, Canada B3H 4R2

ELP, 0000-0001-7811-9288; DL, 0000-0003-1159-3828

The timing of reproduction influences key evolutionary and ecological processes in wild populations. Variation in reproductive timing may be an especially important evolutionary driver in the marine environment, where the high mobility of many species and few physical barriers to migration provide limited opportunities for spatial divergence to arise. Using genomic data collected from spawning aggregations of Pacific herring (*Clupea pallasii*) across 1600 km of coastline, we show that reproductive timing drives population structure in these pelagic fish. Within a specific spawning season, we observed isolation by distance, indicating that gene flow is also geographically limited over our study area. These results emphasize the importance of considering both seasonal and spatial variation in spawning when delineating management units for herring. On several chromosomes, we detected linkage disequilibrium extending over multiple Mb, suggesting the presence of chromosomal rearrangements. Spawning phenology was highly correlated with polymorphisms in several genes, in particular *SYNE2*, which influences the development of retinal photoreceptors in vertebrates. *SYNE2* is probably within a chromosomal rearrangement in Pacific herring and is also associated with spawn timing in Atlantic herring (*Clupea harengus*). The observed genetic diversity probably underlies resource waves provided by spawning herring. Given the ecological, economic and cultural significance of herring, our results support that conserving intraspecific genetic diversity is important for maintaining current and future ecosystem processes.

## 1. Introduction

The conservation of genetic diversity is central to management efforts in many species [1,2], though the functional significance of such diversity for ecologically relevant traits is often unknown. One such trait, the timing of reproduction, underlies key evolutionary and ecological processes in the wild because it mediates gene flow [3] and thus can contribute to population

divergence and speciation [4]. At the ecosystem scale, diversity in reproductive timing reduces the risk of recruitment failure caused by unfavourable environmental conditions [5] and contributes to ecological portfolio effects [6]. Furthermore, resource waves (i.e. trophic resources that are locally abundant for short periods of time) associated with spatial variation in reproductive timing prolong foraging opportunities for mobile consumers [7].

Although the evolutionary and ecological significance of reproductive timing is widely recognized, little is known about its genetic basis. What is known stems largely from plants [8] and insects [9,10], even though the evolutionary effects of allochrony (i.e. differences in timing) may be even more pronounced in marine species. Reproductive allochrony is probably an important evolutionary driver in the marine environment, where the high mobility of many species and few physical barriers to migration reduce opportunities for spatial genetic divergence to arise. Furthermore, the genetic basis of reproductive timing has important ecological implications since the temporal matching between larval fish emergence and plankton production determines larval survival and recruitment success in many temperate marine species [11–13].

Reproductive allochrony may be particularly important in explaining the population structure and biocomplexity of Pacific and Atlantic herring (*Clupea pallasii*, *Clupea harengus*), estimated to have diverged 2.2 Ma [14]. Spawn timing varies across broad latitudinal gradients, but there is also variation within narrow geographical regions for both species [15], and populations with distinct spawning seasons (e.g. winter versus spring spawners in Pacific herring or spring versus autumn spawners in Atlantic herring) are genetically distinguishable [16,17]. The consistency of spawn timing within populations [18] supports the long-standing hypothesis that reproduction and larval emergence are synchronized with cycles of marine productivity [11].

On the west coast of North America, Pacific herring migrate to the nearshore environment in winter and spring to spawn on intertidal and subtidal marine vegetation [15]. Sexually mature fish gather near spawning grounds several weeks or months before spawning [15] and quickly disperse after reproducing. The movement and distribution of Pacific herring outside of the spawning season are poorly understood, but there is evidence from contaminants [19] and stable isotopes [20] that some populations migrate offshore to feed while others reside in coastal waters and estuaries. Mark-recapture studies [21] show that Pacific herring display fidelity to relatively broad geographical areas.

During the spawning season, Pacific herring support commercially and culturally important fisheries. Commercial fisheries harvest eggs from sexually mature fish prior to spawning, while indigenous fisheries primarily harvest eggs that have been deposited on vegetation (and some adult herring are taken for subsistence). The fisheries are managed spatially, though the geographical extent of spatial units varies by country and administrative area (i.e. state or province). In Washington State, spawning biomass is estimated for individual spawning areas and a limited bait fishery harvests herring outside the spawning season [22]. In British Columbia, spawning biomass is estimated for five major and two minor stocks and used to set annual quotas for the commercial fishery [23]. In Alaska, fisheries are managed as regulatory stocks that spawn on specific beaches and coastlines, though regulations vary regionally [24]. In general,

stocks combine multiple local spawning aggregations, many of which have cultural and economic significance for indigenous groups [25,26]. This sets up a conflict between resource users when local spawning aggregations decline in abundance or collapse, as spatially constrained groups (e.g. indigenous fishers) are affected more severely than highly mobile industrial fishing fleets [27]. Furthermore, spatial management schemes may not account for temporal population structure, and declines in spawn timing diversity [28] may impact the resilience of this resource [29].

The temporal and spatial differentiation in both Atlantic and Pacific herring may be surprising given evidence for some straying between spawning locations [21] as well as spawning seasons [30]. Nevertheless, in species with high connectivity, chromosomal rearrangements may facilitate local adaptation [31–33]. Research on Atlantic herring has identified genomic regions associated with spawning season [14,34], some of which are located on a supergene maintained by a chromosomal inversion [35]. It is currently unknown whether these same genomic regions contribute to the fine-scale diversity in reproductive phenology of Pacific herring. If so, it would demonstrate a shared genetic basis of reproductive allochrony that either predates speciation or that arose in parallel between Atlantic and Pacific herring. In this study, we used genomic data collected from spawning aggregations of Pacific herring across 1600 km of coastline to: (i) test whether temporal and spatial differences in reproduction drive genetic differentiation in herring, (ii) estimate the temporal stability of spawn timing from spawner surveys, (iii) compare genomic regions most highly correlated with spawn timing with those found in Atlantic herring, and (iv) test whether genomic regions correlated with divergent spawn timing are associated with chromosomal rearrangements.

## 2. Results and discussion

We sampled 1104 Pacific herring from 23 different spawning aggregations across 1600 km on the Pacific Coast of North America (figure 1*a*). Spawn timing at these sites ranged from January to June (electronic supplementary material, figure S1 and table S1). Restriction site-associated DNA (RAD) sequencing was used to genotype each sample at 6718 polymorphic loci that aligned to the Atlantic herring genome (see the electronic supplementary material for additional details). After filtering for missing data, each individual sample had an average read depth of 75× at these loci.

We tested whether subtle differences in spawn timing (in the range of weeks to months) and geographical distance influence the genetic population structure of Pacific herring. A principal components analysis revealed differentiation between populations spawning at different times and in different geographical regions (electronic supplementary material, figure S2), and the first two principal components were highly significant using the Tracy-Widom statistic ($p < 0.0001$). In a subsequent discriminant analysis of principal components (DAPC) (figure 1*b*), the first discriminant axis explained 38% of the retained variance and separated herring spawning in May and June from all other individuals, while the second discriminant axis summarized 19% of the retained variance and separated samples along a latitudinal and temporal gradient. A pattern of isolation by time [3] was detected (Mantel $r = 0.51$, Mantel $p \leq 0.00099$; figure 1*c*) despite low overall levels of genetic differentiation (global $F_{ST} = 0.014$), indicating limited

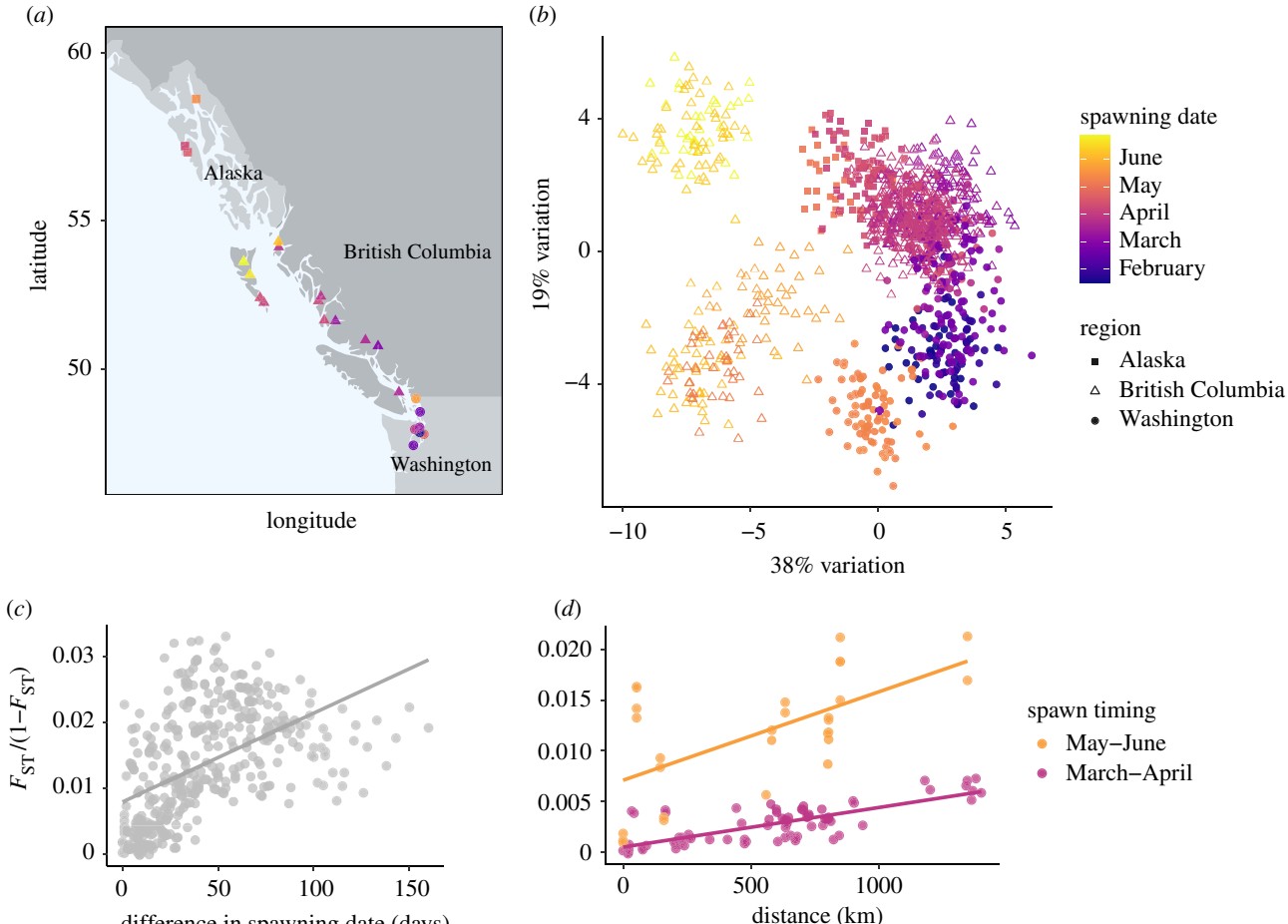

**Figure 1.** Genetic population structure in Pacific herring. (*a*) Map of all sampling locations. The colour of each circle depicts the date of sampling an aggregation of spawning or sexually mature fish. (*b*) DAPC. Each point represents an individual herring. (*c*) Isolation by time in Pacific herring. Linear regression of pairwise $F_{ST}$ to the number of days separating sampling events at each location (Mantel $r = 0.51$, Mantel $p = 0.00099$). (*d*) Isolation by distance in Pacific herring. Points represent pairwise comparisons among sampling locations within a specific spawning period (orange points: May–June spawners; purple points: March–April spawners). Linear regression of pairwise $F_{ST}$ to the geographical distance separating sampling locations is estimated separately for each spawning period (May–June spawners: Mantel $r = 0.61$, Mantel $p = 0.00099$; March–April spawners: Mantel $r = 0.77$, Mantel $p = 0.00099$). (Online version in colour.)

gene flow between herring that spawn at different times. This isolation by time relationship was even stronger after we removed the highly divergent May–June spawners from the analysis (Mantel $r = 0.70$, Mantel $p = 0.00099$). Additionally, there was also evidence of isolation by distance [36] within each specific spawning group (May–June spawners: Mantel $r = 0.61$, Mantel $p \leq 0.00099$; March–April spawners: Mantel $r = 0.77$, Mantel $p \leq 0.00099$; figure 1*d*).

A model-based Bayesian analysis of population structure [37] estimated the highest posterior probability for a model of population structure that included information on spawn timing (posterior probability = 0.96), followed by a model that incorporated both information on spawn timing as well as geographical distance between spawning sites (posterior probability = 0.04). Pacific herring population structure is therefore primarily driven by spawn timing, with limited spatial dispersal within a particular spawning period. As both spawn timing and allele frequencies of collections from different spawning groups were stable over decades (electronic supplementary material, figure S3 and figure S4), genetic differentiation correlated with phenology appears to be a major determinant of population structure in Pacific herring. The interannual consistency of allele frequencies within a particular spawning population has also been observed in Atlantic herring [38], and demonstrates that populations exhibit seasonal and geographical fidelity to spawning sites across

multiple generations. Our results support the hypothesis that Pacific herring home back to their natal spawning site. This is, to our knowledge, the first observation of such fine-scale genetic structure in Pacific herring and it is notable, given the very large census population sizes and high mobility characteristic of these pelagic fish [21]. Our results highlight the importance of considering both seasonal and spatial variation when delineating management units for marine species.

As genome-wide differentiation was low across all populations (global $F_{ST} = 0.014$), it provided a good foundation for investigating selective differentiation associated with seasonal reproduction and spawn timing diversity. Many temperate fishes initiate reproduction in response to seasonal changes in day length (i.e. photoperiod, [39]); in our study system, the photoperiod at which spawning occurred ranged from 8 h in January to 17 h in June. We evaluated whether any single nucleotide polymorphisms (SNPs) were correlated with spawning photoperiod using a Bayesian framework [40] that accounts for the covariance of alleles owing to hierarchical population structure and sampling error. We also conducted a principal component analysis (PCA) to identify SNPs that were outliers in regards to overall population differentiation [41]. There were 116 SNPs that were both PCA outliers and strongly correlated with spawning photoperiod ($\log_{10}$ Bayes factor > 2; figure 2*a,b*). These SNPs were distributed across 17 different chromosomes, suggesting that spawn timing is a

*Proc. R. Soc. B* **288**: 20202398

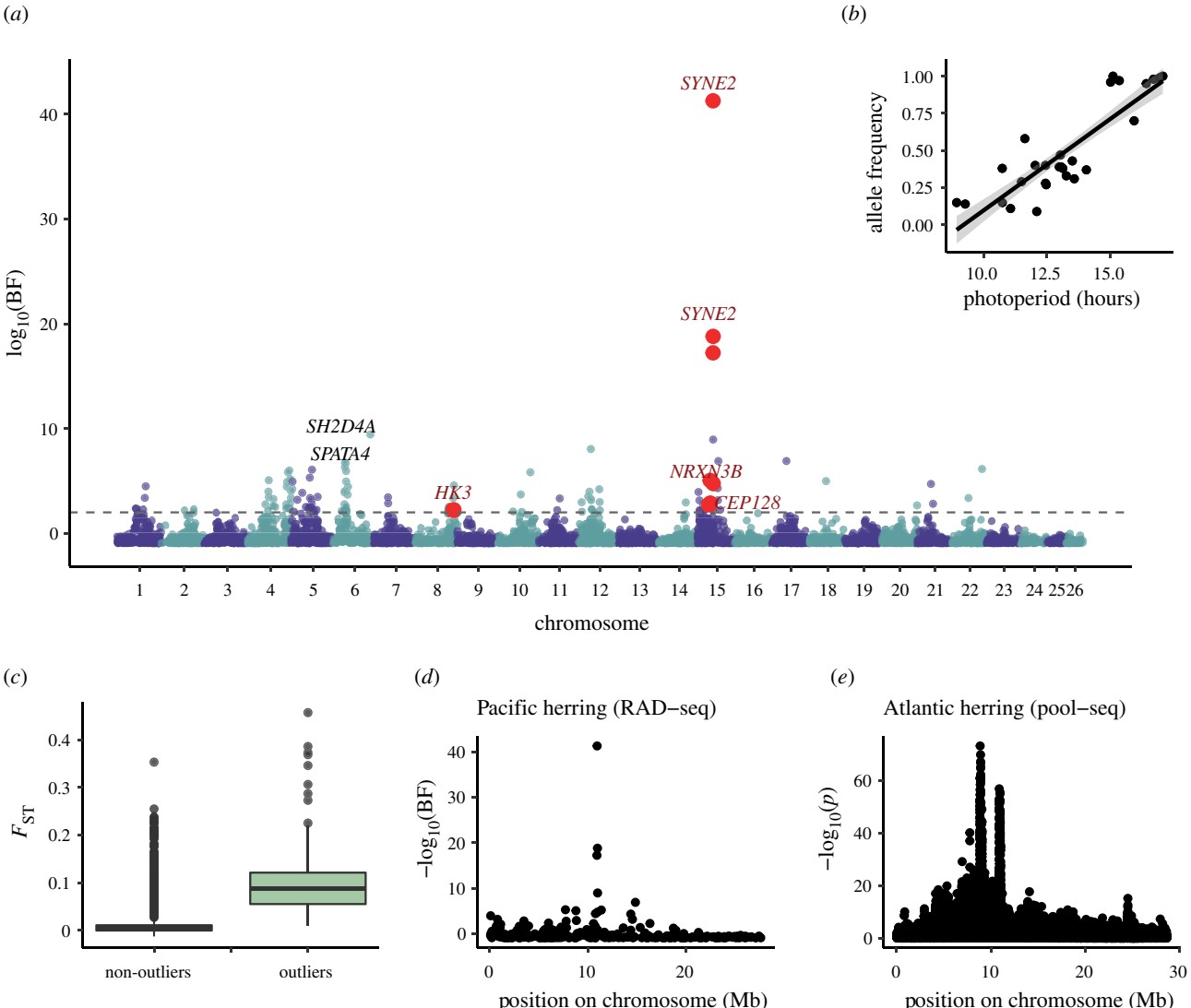

**Figure 2.** Loci correlated with spawn timing in herring. (*a*) SNPs correlated with the photoperiod of reproductive timing in Pacific herring. Points show strength of correlation between the allele frequency of an SNP and photoperiod; points above the dotted grey line are highly correlated with photoperiod (log$_{10}$ Bayes factor (BF) > 2). Red points highlight Pacific herring SNPs that are highly correlated with photoperiod and are also within 5000 bp of an SNP that is associated with spawn timing in Atlantic herring. These shared outlier loci are labelled with the name of the gene they are found within. (*b*) Regression of allele frequency to photoperiod for the SNP most highly correlated with photoperiod in Pacific herring (within *SYNE2*; $F_{ST} = 0.36$; $R^2 = 0.74$, *p*-value = $2.7 \times 10^{-9}$, log$_{10}$ BF = 41). (*c*) $F_{ST}$ distribution of loci correlated with spawn timing in Pacific herring (outliers = loci whose log$_{10}$ BF > 2). (*d*) Strength of correlation with photoperiod in Pacific herring on chromosome 15. The peak of correlation occurs approximately 10.9 Mb and contains five SNPs in *SYNE2*. (*e*) Strength of correlation with spawn timing in Atlantic herring on chromosome 15. At approximately 10.9 Mb, SNPs highly associated with spawn timing are found within *SYNE2*. (Online version in colour.)

polygenic trait. Differentiation over all samples at these shared outlier loci was an order of magnitude higher than at other loci (figure 2*c*; mean $F_{ST}$ of shared outlier loci = 0.11; mean $F_{ST}$ of other loci = 0.01).

A subset of these outlier loci (*n* = 44; 36%) were found within chromosomes 8, 12 and 15 which had high linkage disequilibrium (LD) extending over several Mb (figure 3*a,b*; electronic supplementary material, figures S7 & S8). By contrast, LD decayed very quickly on most other chromosomes (electronic supplementary material, figure S9). Network analysis of LD [42] showed that 34 of these shared outlier loci formed networks of high intrachromosomal LD on chromosomes 8, 12 and 15, even though some of the loci were separated by millions of base pairs. These patterns of LD are consistent with the presence of large haplotype blocks caused by processes that suppress recombination over extended genomic distances, such as large chromosomal rearrangements [43,44] or linked selection [45]. We used a

PCA-based approach to identify groups of individuals with distinct genotypes on these chromosomes and found that PCA clustered individuals into three distinct groups (figure 3*c*; electronic supplementary material, figures S7 and S8). This pattern is characteristic of chromosomal rearrangements such as inversions [46], where the distinct PCA clusters correspond to homo- and heterokaryotypes. Consistent with this hypothesis, we observed higher heterozygosity for individuals in the intermediate PCA cluster (figure 3*d*). Within a particular spawning site, tests for Hardy-Weinberg equilibrium (HWE) indicated that individuals were approximately randomly mating with regards to the putative inversions (6% of tests were statistically significant at $\alpha < 0.05$). Hierarchical analysis of molecular variance (AMOVAs) showed that the frequency of putative inversion genotypes (figure 3*e*) differed between groups with different spawning phenology (January–February versus March–April versus May–June spawners). When all spawning groups were included in the analysis, statistically

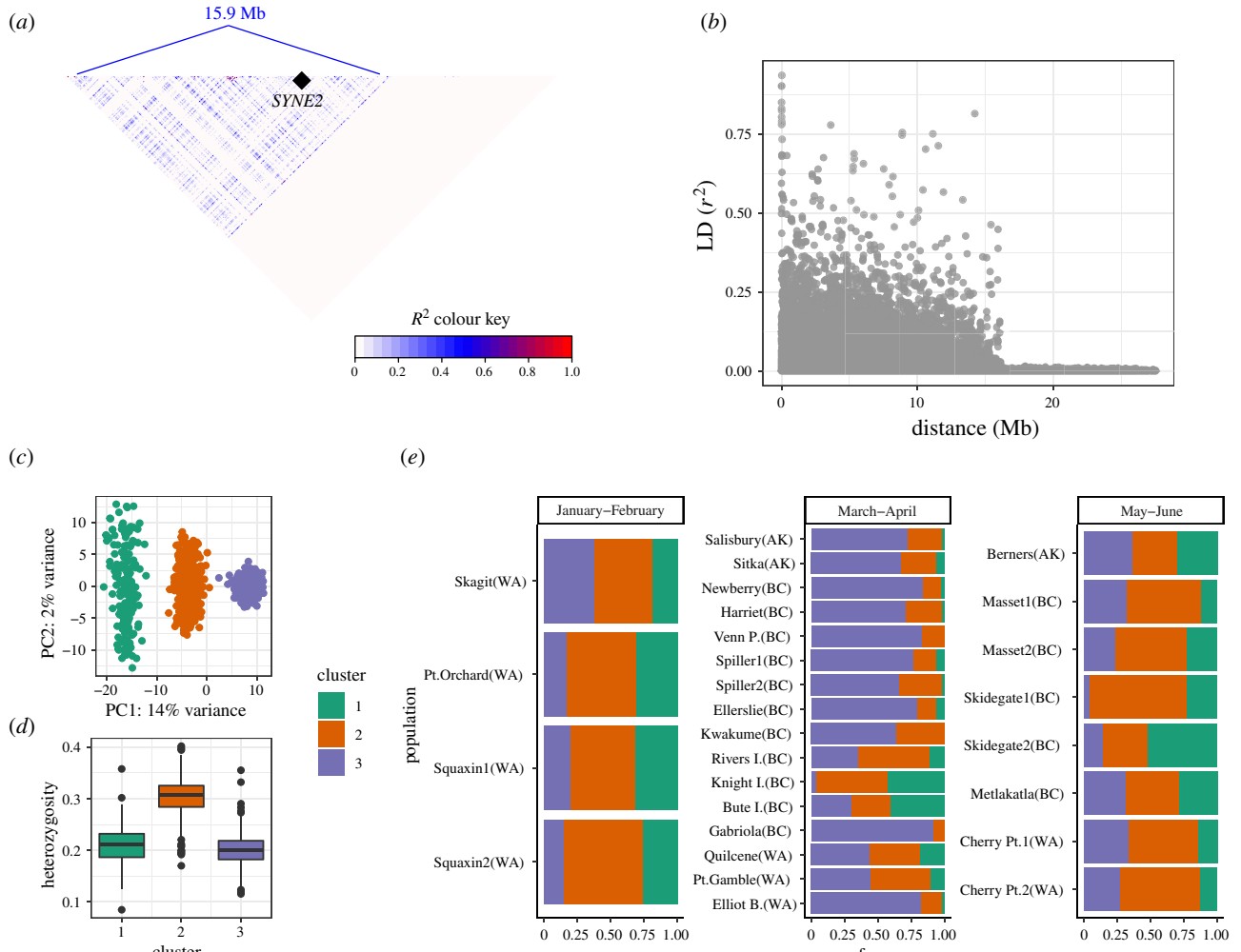

**Figure 3.** Patterns of linkage disequilibrium (LD) on chromosome 15 in Pacific herring. (*a*) Heat map of pairwise LD over a distance of 15.9 Mb on a chromosome whose total size is 28.71 Mb. The relative position of *SYNE2* is marked by a black diamond. (*b*) Decay of LD as a function of the distance separating two loci. (*c*) PCA of chromosome 15 loci using all individuals. (*d*) Observed individual heterozygosity in each PCA cluster. (*e*) Allele frequency distributions for putative inversion genotypes across different sampling locations. Bars represent the proportion of individuals belonging to a PCA cluster (indicated by colour). Sampling locations are labelled with their geographical region in parentheses (WA, Washington; BC, British Columbia; AK, Alaska). (Online version in colour.)

significant ($p < 0.0001$) differences in genotype frequency were observed across all hierarchical levels and chromosomes (electronic supplementary material, table S3). When only January–February and March–April spawners were included, there was no difference in the frequency of putative inversion genotypes on chromosomes 12 and 15 (electronic supplementary material, table S3), suggesting that these spawning groups may share more recent ancestry with one another than with March–April spawners.

In Atlantic herring [35], there is an extensive LD block on chromosome 12 that is maintained by an inversion and exhibits clinal patterns in allele frequency from north to south. It was hypothesized that this genomic region may act as a supergene involved in ecological adaptation to water temperature during gonadal development and/or spawning [35]. In other systems, inversions have been associated with local adaptation and divergence with regards to migratory behaviors (Atlantic cod, [47]), ecotypes (dune sunflower, [48]), reproductive strategies (ruff, [49]) and life-history strategies (steelhead/rainbow trout, [50]).

Outlier SNPs with the strongest correlations to spawning photoperiod ($\log_{10}$ Bayes factor >10; figure 2*d*) were located on chromosome 15 and mapped to the mRNA encoding SYNE2, a nuclear membrane protein that (among other functions) influences the position of retinal photoreceptors early

in vertebrate development [51,52]. All of the SNPs within *SYNE2* that were strongly correlated with spawning photoperiod ($n = 5$) were found within the coding sequence of this gene. The SNP with the highest correlation to spawning photoperiod ($\log_{10}$ Bayes factor = 41) was a non-synonymous missense variant (serine to histidine), which is predicted to have a moderate effect on the protein structure. The other outlier SNPs in *SYNE2* represented intron variants or splice region variants. Additional genomic regions that were highly associated ($\log_{10}$ Bayes factor > 5) with spawning photoperiod in Pacific herring and may play a role in reproductive pathways include two loci on chromosome 6: *SH2D4A*, a locus with oestrogen receptor binding activity [53] and *SPATA4*, a locus associated with spermatogenesis [54]. Together, these associations provide support for a mechanistic link between the population structure, spawn timing, and the photoperiodic regulation of reproduction in Pacific herring.

Such mechanistic links provide an opportunity to test for parallel evolution—the evolution of mutations in independent lineages that result in phenotypic similarities [55]. In Atlantic herring, a specific subset of genes related to the photoperiodic regulation of reproduction are highly associated with spawn timing in populations from both sides of the Atlantic Ocean [34], but as haplotypes are shared among populations, they may have arisen from standing

genetic variation in the ancestral population or gene flow across the Atlantic. By contrast, under parallel evolution, one would expect to observe independent neutral SNPs linked to the same functional mutation if adaptive alleles arose independently. We first investigated whether SNPs in the same genomic regions were correlated with reproductive phenology in Atlantic and Pacific herring, and found SNPs within four genes (*SYNE2, NRXN3B, CEP128, HK3*) that were highly correlated with reproductive timing variation in both species (figure 2*a*), suggesting that similar genes may underlie highly complex traits such as the timing of reproduction. Furthermore, it is likely that these loci are within chromosomal rearrangements or subject to linked selection in Pacific herring, as LD network analysis showed that SNPs within three of these genes (*SYNE2, NRXN3B, CEP128*) were found in networks of high LD on chromosome 15, while the SNP within *HK3* was found in networks of high LD on chromosome 8.

SNPs in *SYNE2* were among the most highly associated with spawn timing in both species (figures 2*d,e*), suggesting a strong effect of that gene on reproductive timing. However, specific SNPs in Pacific herring differed from those in Atlantic herring; the five SNPs within *SYNE2* found in Pacific herring were monomorphic in Atlantic herring. This pattern suggests that species-specific polymorphisms arose independently and may indicate parallel evolution of varied reproductive phenologies across sister species. Future studies comparing the two species by conducting targeted sequencing in this genomic region should be able to rigorously test this hypothesis. Interestingly, winter and spring spawning populations of Atlantic cod also exhibit large allele frequency differences [56] in *SYNE2*, providing additional evidence that this gene or genomic region plays a role in reproductive phenology in a variety of fish taxa.

Our results indicate that genetic variation underlying spawn timing structures herring populations in time and space. Given the crucial role of forage fishes in marine ecosystems of converting primary and secondary production to energy for higher trophic levels [57,58], it is necessary to consider how the observed patterns of genetic variation and population structure interact with other ecological processes. Phenological variation provides an important buffer against environmental disturbance, especially in species central to ecosystem functioning. Genetic variation in reproductive timing is probably the basis of the prolonged resource wave that spawning herring provide to coastal environments as they aggregate to deposit energy-rich eggs [7,59]. In addition, limited gene flow between temporally and/or spatially separated spawning aggregations probably contributes to the independent population dynamics and portfolio effects that have been observed in Pacific herring [29].This spatio-temporal population structure may also increase the vulnerability of Pacific herring populations to unintentional overexploitation by fisheries at fine spatial scales [27]. Undetected population structure within management units can lead to 'cryptic collapses' [27] that may reduce portfolio effects maintaining population stability at the species level [6]. Temporal population structure in spatially managed species may be particularly vulnerable if fisheries targeting, for example, early spawners also accidentally exploit smaller late-spawning populations. Such bycatch could be quantified and considered in assessment by genetic population assignment using some of the highly discriminatory loci [60] identified in this study.

Identifying the genetic basis and environmental drivers of variation in reproductive phenology is also important for predicting impacts of climate change and other anthropogenic disturbances on marine food webs. It is currently unknown how photoperiod interacts with other environmental cues (such as sea surface temperature or tidal cycle) to influence the timing of reproduction in herring. Recent studies of marine phenology indicate that climate change and the warming of surface waters are causing shifts towards earlier larval emergence in some pelagic fishes [61,62], a phenomenon that could lead to trophic mismatches and increased recruitment variability. Furthermore, increasing sea surface temperatures could influence metabolic processes during early life-history stages [63,64], thus leading to emergent effects that impact survival. However, species whose reproductive timing has a strong genetic basis and is driven by consistent environmental stimuli such as day length may also be vulnerable to climate change because of a reduced ability to shift reproductive timing in response to changing thermal conditions.

Current fishery management practices [65] as well as global climate change [66] may irrevocably reduce genetic diversity in Pacific herring before its significance is fully recognized, thereby reducing the ability of species to adapt to future environmental conditions. Given that forage fishes like Pacific herring are foundational to coastal food webs [58], fisheries, and the livelihoods and cultures of coastal indigenous communities [25,26], diversity loss will have far-reaching consequences for associated social-ecological systems [67]. Therefore, fishery and habitat management practices should strive to maintain existing spatial and temporal population diversity and mitigate trends in spawn timing compression [28], as this diversity may enhance species-level resilience to environmental variation and climate change.

## 3. Methods

### (a) Sample collection
Approximately 48 herring were collected from each of 23 distinct spawning sites spanning 1600 km of the Pacific northwest coast of North America (electronic supplementary material, figure S1 and table S1). All but one of these sample collections consisted of sexually mature herring caught in pre-spawning aggregations (before 2013) or during active spawning events (after 2013) using nets or hook-and-line fishing gear. Late-stage eggs were collected at one site (electronic supplementary material, table S1). Five different locations were sampled twice across different years (range of years separating spawning events = 1–15 years); we hereinafter refer to these samples as 'temporal replicates'. All herring ($n = 1344$) were collected by state, federal, or tribal biologists, or permitted subsistence fishers. Subsampled tissue was individually placed in 100% ethanol.

### (b) DNA sequencing and genotyping
We followed the protocol of Etter *et al.* [68] to prepare DNA for RAD sequencing (see the electronic supplementary material for details). DNA libraries were standardized to a concentration of 10 nM and pooled such that 96 individuals were sequenced per lane of an Illumina HiSeq 4000 at the University of Oregon Genomics Core Facility. The resulting sequences were single-end and had a length of 100 bp.

We used STACKS v. 1.46 [69] to analyse sequencing data and genotype samples. Loci were filtered for missing data, minor allele frequency and potential paralogues (see the electronic

supplementary material for details). As a final quality control measure and in the absence of a genome assembly for Pacific herring, we aligned loci to the chromosome-level assembly of the Atlantic herring genome [35] using Bowtie2 v. 2.2.6 (–sensitive alignment) [70] and only retained loci that aligned with a mapping quality greater than or equal to 20. Individual herring were assessed for missing data and removed if they had more than 10% missing genotypes. We screened individuals for intraspecific DNA contamination by calculating individual multilocus heterozygosity ($H_I$) and comparing it to a set of clean reference samples [71]. Individual samples were removed from the dataset if they had $H_I$ values higher than clean reference samples ($H_I > 0.32$), as in Petrou et al. [71].

## (c) Population structure

Pairwise $F_{ST}$ [72] between sample collections, $F_{IS}$ within each sample collection, and per-locus $F_{ST}$s were calculated using the R package hierfstat v. 0.04.22 [73]. We visualized population structure using a PCA as implemented by smartPCA in Eigensoft v. 7.2 [74]. As input data, we used only biallelic SNPs that had been pruned for linkage disequilibrium (–indep-pairwise 100 10 0.1) using plink v. 1.9 [75]. We assessed variance between sampling locations using a DAPC with adegenet v. 2.1.3 [76]. Groups were defined a priori based on sampling location and collection year and the optim.a.score function was used to identify the number of principal components to retain in the analysis.

To test whether the timing of reproduction affected genetic differentiation (isolation by time; 3), we conducted a linear regression of linearized pairwise $F_{ST}$ ($F_{ST}/(1 - F_{ST})$) to the number of days separating sampling events. The statistical significance of isolation by time was evaluated using a Mantel test (Pearson's product moment correlation; 10 000 permutations) as implemented in the R package vegan v. 2.5.6 [77]. We also tested for isolation by distance [36] by conducting a linear regression of linearized pairwise $F_{ST}$ to the geographical distance separating sample collections. The shortest distance (as the crow flies) separating two sampling locations was calculated using the R package geosphere v. 1.5.10 [78], and the statistical significance of isolation by distance was evaluated as described above for time.

We assessed the relative importance of reproductive timing and geographical distance in structuring Pacific herring populations by conducting a hierarchical Bayesian analysis of population structure with the program GESTE v. 2.0 [37] (see the electronic supplementary material for details).

## (d) Outlier loci and loci correlated with spawn timing

We identified SNPs that were outliers to overall patterns of population differentiation in our samples using PCA, as implemented in the R package pcadapt 4.3.3 [41]. We evaluated whether SNPs were associated with spawning photoperiod using bayenv2, which accounts for the covariance of alleles owing to hierarchical population structure and sampling error [40] (see the electronic supplemental materials for details).

## (e) Linkage disequilibrium

We estimated linkage disequilibrium (LD) within each chromosome using the R package genetics v.1.3.8 [79]. LD network analysis [42] was used to identify networks of loci in high LD (parameters used: $E_{min} = 20$, $\varphi = 7$). Because several chromosomes contained loci in strong LD ($r^2 > 0.5$) over extended distances (greater than 1 Mb), we conducted intrachromosomal PCAs in adegenet to identify whether individuals clustered in the distinct 'three stripe' pattern characteristic of chromosomal rearrangements such as inversions [46] (see the supplementary material for details). To assign individuals to a putative

chromosomal inversion, we applied k-means clustering to the first two eigenvectors of the PCA using the R function kmeans.

We calculated observed heterozygosity for individuals in each of the clusters identified by PCA. Our expectation was that the central cluster, representing individuals that are heterozygous for the putative inversion, would have high heterozygosity relative to the other clusters. We calculated genotype frequencies for the putative inversions in each population and tested for statistically significant differences in allele frequencies using an AMOVA with Arlequin v. 3.5.2 [80]. We evaluated whether individuals were randomly mating in regards to inversion genotype by testing for deviations for HWE within each population using the exact test (1000 Monte Carlo permutations of alleles) as implemented in the R package pegas v. 0.13 [81].

## (f) Comparison to Atlantic herring data

We investigated whether the same genomic regions and SNPs were correlated with reproductive timing in both Atlantic and Pacific herring, which are sister species that diverged approximately 2 Ma and now inhabit different oceans. Atlantic herring SNPs correlated with spawn timing were previously published in Lamichhaney et al. [34] and Fuentes-Pardo et al. [82]. Using the comprehensive list of spawn timing loci compiled by Fuentes-Pardo et al. [82], we assessed whether SNPs correlated with spawn timing in Pacific herring were proximate to SNPs or genes correlated with spawn timing in Atlantic herring (see the electronic supplementary material for details).

Data accessibility. Sequencing data reported in this study have been deposited in the National Center for Biotechnology Information Sequence Read Archive (https://www.ncbi.nlm.nih.gov/sra) and are accessible through BioProject IDs PRJNA508972, PRJNA669540 and PRJNA669068. Genotyping data generated in this project can are available from the Dryad Digital Repository: https://doi.org/10.5061/dryad.80gb5mkpj [83]. Bioinformatic scripts used to analyse sequencing and genotypic data are available on GitHub (https://github.com/EleniLPetrou/pacific_herring_RADseq).

Authors' contributions. This project was designed by L.H., E.L.P., T.J.P., D.Y., M.L.M. and D.E.R. Data analysis was conducted by E.L.P., A.P.F.P., L.A.R., M.O., T.S., I.J.H. and C.T. The manuscript was written by E.L.P., L.H., A.P.F.P. and L.A.R. All authors provided comments on drafts of the manuscript and approved the final version.

Competing interests. We declare we have no competing interests.

Funding. This work was funded in part by a grant from Washington Sea Grant, University of Washington, pursuant to National Oceanic and Atmospheric Administration (award no. NA14OAR4170078, project no. R/HCE-3). The views expressed herein are those of the authors and do not necessarily reflect the views of NOAA or any of its sub-agencies. Additional support was provided by the Natural Sciences and Engineering Research Council of Canada Strategic Partnership Grant (no. 447247) and a US National Science Foundation award (no. 1203868). E.L.P. received additional support from the University of Washington Program on Ocean Change Integrative Graduate Education and Research Traineeship, funded by the NSF award no. 1068839. We thank the Tula Foundation for supporting our fieldwork at the Hakai Institute.

Acknowledgements. We thank the following people and organizations for sharing expert knowledge and samples: Dana Lepofsky (Simon Fraser University); Kelly Brown, Mike Reid and Davey Wilson (Heiltsuk Integrated Resource Management Division); Russ Jones (Haida Fisheries Program); Lynn Lee, Leandre Veignault and the Lavois family; Kurt Stick, Adam Lindquist and Wade Smith (Washington Department of Fish and Wildlife); Terry Beacham, Jaclyn Cleary and Matthew Grinnell (Department of Fisheries and Oceans Canada); Kira Krumhansl; Margot Hessing-Lewis (Hakai Institute); Sharon Wildes (NOAA Juneau); Chad Ormond (Q'ulhanumutsun Aquatic Resources Society); Kyle Rosendale, Jeff Feldpausch and Mike Smith (Sitka Tribe of Alaska). We thank Molly Phillips for assistance in the laboratory, and Dan Drinan, Natalie Lowell, Mary Fisher and Charlie Waters for bioinformatic support and comments to early versions of this manuscript.

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
