## [Peer Review File · Proceedings of the Royal Society B: Biological Sciences]

Review History

RSPB-2020-2398.R0 (Original submission)

Review form: Reviewer 1

Recommendation

Accept with minor revision (please list in comments)

Scientific importance: Is the manuscript an original and important contribution to its field?

Good

General interest: Is the paper of sufficient general interest?

Good

Quality of the paper: Is the overall quality of the paper suitable?

Excellent

Is the length of the paper justified?

Yes

Should the paper be seen by a specialist statistical reviewer?

No

Do you have any concerns about statistical analyses in this paper? If so, please specify them explicitly in your report.

No

It is a condition of publication that authors make their supporting data, code and materials available - either as supplementary material or hosted in an external repository. Please rate, if applicable, the supporting data on the following criteria.

Is it accessible?

No

Is it clear?

Yes

Is it adequate?

Yes

Do you have any ethical concerns with this paper?

No

Comments to the Author

The manuscript "Functional genetic diversity in an exploited marine species and its relevance to fisheries management" by Eleni Petrou and colleagues uses RAD sequencing of Pacific herring samples to investigate whether spatial and temporal shifts of spawning aggregations have an underlying genetic basis. They find that population structure is majorly driven by temporal differences in spawning time followed by spatial distance, identify genes related to light reception possibly influencing photoperiodic regulation of reproduction, and advice fisheries management authorities to take not only spatial, but also temporal population diversity into account.

I read the manuscript with great joy and interest! The manuscript is written in a clear and understandable fashion, it is logically sectioned, the methods are explained in appropriate detail and accompanied by custom scripts on a repository, and the analyses are of high scientific quality. There are no major issues that I could identify, but I have some suggestions that may strengthen the paper even further.

1) Relevance for fisheries management

It may be difficult to find the right balance between phrasing the manuscript as a recommendation for fisheries manager or a broadly interesting topic for scientists. I would leave it up to the authors to decide on which aspect they want to focus, but I would recommend to follow this aim throughout the manuscript.

At the moment the title stresses the relevance to fisheries management, but the Introduction is short on any information about current management strategies for Pacific herring. For example, is all Pacific herring along the coast managed as one stock? Or are there distinct management areas? If so, do these areas lump populations that show asynchrony in spawning time together? Is Pacific herring fished throughout the year or are there restrictions during certain times that could protect one but not the other spawning population? Perhaps management areas could also be included in the map on Figure 1a.

Then, for the Discussion I would recommend to come back to the current management strategy outlined in the Introduction and add a brief paragraph on how the here obtained results could be used to improve this strategy. There is an ongoing debate within many conservation research areas on how to bridge the communication gap between scientists and conservation or management practitioners. This lack of communication and lack of mutual understanding was also highlighted in an article about the role of genomics to secure the future of seafood

(Bernatchez et al. (2017) *Trends Ecol Evol*, 32, 665-680; <https://doi.org/10.1016/j.tree.2017.06.010>).

So, if fisheries management authorities would approach you and ask for advice on how the findings of this project could be applied to the management of Pacific herring or fish populations in general, what would you recommend? Could for example specific SNPs identified in this study be used to genetically monitor the populations? Or identify and control landings of specific populations? Or could spatial management areas be closed for fishing during certain times to protect distinct populations? Perhaps a brief paragraph on this issue and any actions taken by the authors to bridge the gap between scientists and practitioners could be added in order to support the integration of genomic-informed methods into management and raise awareness of this issue.

2) More on the biology

In the Introduction, I also missed some information about the biology and life history of the Pacific herring populations. Do these populations undergo spawning or foraging migrations or do they remain in their local areas? I guess the eggs and larvae drift with the ocean currents, so for populations to stay differentiated throughout time (some generations?) as shown in this manuscript, they have to return to their birth places or the larvae to be trapped in some sort of local recruitment system. Does natal homing exist in herring? Could you add a short paragraph on this?

3) The issue with DAPCs

Discriminant Analysis of Principal Components (DAPC) is not to be used to test for population structure (see e.g., Miller et al. (2020) *Heredity*, 125, 269–280, <https://doi.org/10.1038/s41437-020-0348-2>). Using DAPC with prior groupings (here: sampling location) will always display cluster according to the sampling locations. That is, because the algorithm tests for differences between the a priori specified groups. Having said this, it is valid to ask whether temporal or spatial distance has the larger influence on population structure in the dataset and use DAPC to answer this question.

I suggest to carefully rephrase all sections concerning the DAPC analyses to explicitly state that the intention is not to identify or visualize population structure, but whether temporal or spatial distance has the larger variance in such.

In addition, I would appreciate if a regular principal component analysis (smartPCA, or “STRUCTURE”, or “fineRADstructure”, or a similar analysis) could be added, to visualize the general population structure in the data (perhaps remove the highly linked regions for such analyses). Maybe also the F_{ST} distance matrix (Fig. S113) could be used for this, if it is possible to add for example above the diagonal a heat-plot according to high/low F_{ST} and if samples can be clustered according to the strongest pattern.

4) Outlier SNPs in LD region

Among temporally diverged spawning aggregations, several outlier SNPs were identified in a region on chromosome 15. This is a region with high linkage disequilibrium (LD) that may be a chromosomal rearrangement. The strongest outlier are SNPs in a gene that influences the position of retinal photoreceptors, which could be related to light perception and thus regulate photoperiodic reproduction, leading to spawning events at different times of the year.

If this region is in fact a chromosomal rearrangement, recombination is limited in heterozygous individuals and thus any gene within this region could be under selection. What are the other genes in this region? Is there a common function to these genes (i.e., was a gene ontology (GO) analysis done)? Then, the five SNPs in SYNE2 that are monomorphic in Atlantic herring, are they in the coding region? If so, are these synonymous or non-synonymous substitutions, and do they potentially change the protein structure? If they're not in the coding region or synonymous, they may as well be neutral and do not support parallel selection of this gene.

5) IBD using the shortest distance

For measures of isolation-by-distance (IBD), the shortest distance “as the crow flies” was used, ignoring landmasses and seascapes. Is this a meaningful approach? According to the map, there are many fjords. Possibly the direct line between two locations that are deep in two adjacent

fjords (in the fjord part furthest away from the sea) is closer, than a direct line between the mouth and the end of the fjords, although these latter positions would likely experience more exchange than the two locations in different fjords (I hope it is clear what I mean ;)). There would be another method called “least-cost paths”, taking landmasses and seascape into account (including bathymetric data) that could be used instead, which would probably be a more realistic measurement for the distance (R package MARMAP (Pante & Simon-Bouhet (2013) PLoS ONE 8, e73051; <https://doi.org/10.1371/journal.pone.0073051>).

Minor comments:

L87-90 How much of the variation in spawning time may be plasticity or epigenetic modifications? Is there some literature on that? Perhaps some information could be added.

L93-95 “Recent studies have demonstrated that chromosomal rearrangements can facilitate local adaptation in species characterized by subtle population structure and/or high levels of gene flow (19, 20)” Such studies are not that recent anymore, e.g. the monkey flower paper by Twyford et al. is from 2015. Perhaps replace “recent studies...” with “studies in various species...” and add an example for insects (e.g., mosquito, *Drosophila*, or *Heliconius*). Furthermore, the paper by Kess et al. focuses on the divergence of Atlantic cod across the species' range and not necessarily on adaptation in local populations with subtle population structure and/or high levels of gene flow. A much better fit would be the studies on Atlantic cod in local fjord systems such as Barth et al. (2019) *Mol. Ecol.* 28, 1394-1411 (<https://doi.org/10.1111/mec.15010>).

L 110-112 This phrasing is a bit misleading. The 6,718 loci are used in the final analyses, the original number of genotyped loci was probably much higher.

L 112-114 I first thought each individual was sequenced again. Perhaps rephrase to: “After filtering for missing data, each individual sample had an average read depth of...”

L138-140 Here I wondered about larval drift, spawning/foraging migrations, and natal homing in Pacific herring. After having added something about this in the introduction, you could refer to it here.

L192 “the same genes”, which genes?

L263 Are you sure you used 100% ethanol?

L270 nM should not be italic

L272, 276 Base pair may be abbreviated with bp

L274 For demultiplexing, how many mismatches of the barcodes were allowed? Did you check reads for having the correct cut sites?

L273-304 At all steps, could you add how many loci/SNPs and/or individuals you had before and then after each filtering step or how many were removed, respectively? The default dataset was not filtered for linkage disequilibrium, correct? Could you clarify in which analyses all loci were included and where only unlinked loci were included?

L327 “..incorporating environmental data..”, add “..incorporating environmental data (we used geographic and temporal connectivity)..” and remove this information from L331

L347 only the p in p-value should be italic

L357-359 Using the sampling date? Or the approximate date of spawning? Since some of the samples were collected in pre-spawning condition, the approximate date of spawning may be more appropriate.

L359-362 How much burn-in was excluded?

L376 k in k-means should be italic

L383-385 ...as described before?

L388 remove comma between Pacific herring and sister species

Methods section general (including Supplemental methods): Include version number for each software specified.

References general: There are some references with their title capitalized in title case (e.g., L 469, 509), make sure all titles are small caps (unless names etc.). Also, check that species names are italic (e.g., L469)

Figure 3a (and SI6, 7a): Could the total length of the chromosome and the length of the LD region be added to the Figure?

Figure 3e: Could you indicate which allele frequency differences were identified as significantly

different in the Figure?

Supplemental information:

L16 Please include the demultiplexing step and how many mismatches were allowed
L41-47 “in each sampling location” - merging temporal samples from the same location? How many loci were excluded based on HWE filtering? Were loci also exclude if deviating from HWE in only a single population? Were loci excluded showing both heterozygous excess and homozygous deviation? Selection can of course also cause deviations from HWE and if SNPs in even a single population, not checking for homo-/ or heterozygous deviation were removed with a significance threshold of 0.05, important adaptive variation may have been unintentionally removed.

SI Table 1 Please include the month (or period) each population is spawning and the exact date used in the analyses. Also, the abbreviations in the provided VCF file do not fully match the location names given in this table. Please either add the abbreviations to the table or re-name the individuals in the VCF accordingly.

GitHub: The file “Julian_date_metadata.txt” is not provided as SI data nor on GitHub. Perhaps this file and any other data file that might be needed to reproduce the figures using the provided scripts could be added.

Review form: Reviewer 2

Recommendation

Accept with minor revision (please list in comments)

Scientific importance: Is the manuscript an original and important contribution to its field?

Excellent

General interest: Is the paper of sufficient general interest?

Excellent

Quality of the paper: Is the overall quality of the paper suitable?

Good

Is the length of the paper justified?

Yes

Should the paper be seen by a specialist statistical reviewer?

No

Do you have any concerns about statistical analyses in this paper? If so, please specify them explicitly in your report.

No

It is a condition of publication that authors make their supporting data, code and materials available - either as supplementary material or hosted in an external repository. Please rate, if applicable, the supporting data on the following criteria.

Is it accessible?

Yes

Is it clear?

Yes

Is it adequate?

Yes

Do you have any ethical concerns with this paper?

No

Comments to the Author

The current study aims at studying functional genetic diversity in Atlantic herring, with a focus on reproductive timing. The dataset used here is invaluable, containing over 1100 individuals from 23 spawning locations along the Pacific coast of North America.

Overall the quality of the work is very high and the writing excellent, but I would recommend to reformulate the abstract to better emphasize the implications of the findings for fisheries management. Additionally several full sentences in the beginning of the introduction are exactly the same as in the abstract (l.67-70 and 79-81), so I encourage rewriting either in the abstract or in the introduction.

Another general comment concerns the PCA done with PCAdapt. Here it appears that loci in LD (likely corresponding to inversions here) were kept for the analysis. After reading the manual for PCAdapt, the authors recommend to remove loci in LD for the analysis. What was the justification to keep them in?

l. 282: why the cutoff of 20,000 loci? How many loci were in the database?

l. 282: Why couldn't the authors use all individuals in cstacks? This would probably bias the SNP identification?

l.284-285: why removing loci that had more than 20% missing data in any one sampling location? Because of potential SNP in restriction site or null allele?

l.294: MAF 0.05 across all samples = over 100 occurrences (50 homozygotes or 100 heterozygotes). So it's unlikely that they are genotyping errors. Maybe they shouldn't be removed when trying to find adaptive loci? See Ahrens et al. 2018.

l.298: What parameters for bowtie2?

l. 342: how many loci were biallelic?

l. 370-371: Are the authors talking specifically about inversions here (based on the wording "alternate orientation")? If that's the case, it should be explicit. There are other forms of rearrangements that do not involve orientation changes (although they do have impacts on recombination).

Decision letter (RSPB-2020-2398.R0)

09-Dec-2020

Dear Dr Petrou:

Your manuscript has now been peer reviewed and the reviews have been assessed by an Associate Editor. The reviewers' comments (not including confidential comments to the Editor) and the comments from the Associate Editor are included at the end of this email for your reference. As you will see, the reviewers and the Editors have raised some concerns with your manuscript and we would like to invite you to revise your manuscript to address them.

Research ethics:

Use of animals and field studies:

It is a condition of publication that you make available the data and research materials supporting the results in the article. Please see our Data Sharing Policies (<https://royalsociety.org/journals/authors/author-guidelines/#data>). Datasets should be deposited in an appropriate publicly available repository and details of the associated accession number, link or DOI to the datasets must be included in the Data Accessibility section of the article (<https://royalsociety.org/journals/ethics-policies/data-sharing-mining/>). Reference(s) to datasets should also be included in the reference list of the article with DOIs (where available).

All supplementary materials accompanying an accepted article will be treated as in their final form. They will be published alongside the paper on the journal website and posted on the online

figshare repository. Files on figshare will be made available approximately one week before the accompanying article so that the supplementary material can be attributed a unique DOI. Please try to submit all supplementary material as a single file.

Please submit a copy of your revised paper within three weeks. If we do not hear from you within this time your manuscript will be rejected. If you are unable to meet this deadline please let us know as soon as possible, as we may be able to grant a short extension.

Best wishes,
Dr Daniel Costa
mailto:proceedingsb@royalsociety.org

Associate Editor

Comments to Author:

This study has now been thoroughly reviewed by two experts in the field. I agree with both Referees that an MS convincingly demonstrating how the genetic structure of Pacific Herring populations is driven by temporal differences in spawning time followed by spatial distance is a significant finding of broad interest to the community, with implications for the role and evolutionary influence of temporal population diversity in natural populations. Thus, the study has the potential to be of broad interest to PRSB.

Nonetheless, the Referees draw attention to certain caveats in the approach and writing that should be addressed to further strengthen the manuscript. First, recent data has drawn attention to some of the limitations of DAPC (Referee 1). The Referee makes an excellent suggestion to clarify that DAPC is being used to assess variance in temporal and spatial distance with this method. I agree additional analyses here (e.g., smartPCA), removing highly linked regions please, would help support the conclusions. In addition, the notion that the results will be applicable to fisheries management falls a little flat and is somewhat unconvincing as presented. Both Referees outline specific scenarios that could be addressed to strengthen this aspect. Overall, I agree that the issue is an important one. I am also very curious about MARMAP to address a more meaningful representation of distance in the IBD analysis (Referee 1). Both Referees make several other important observations and suggestions that should be carefully considered in the revision. For all of these reasons, I am recommending that the MS be revised following these helpful reviews.

Reviewer(s)' Comments to Author:

Referee: 1

Comments to the Author(s)

The manuscript "Functional genetic diversity in an exploited marine species and its relevance to fisheries management" by Eleni Petrou and colleagues uses RAD sequencing of Pacific herring samples to investigate whether spatial and temporal shifts of spawning aggregations have an underlying genetic basis. They find that population structure is majorly driven by temporal differences in spawning time followed by spatial distance, identify genes related to light reception possibly influencing photoperiodic regulation of reproduction, and advice fisheries management authorities to take not only spatial, but also temporal population diversity into account.

I read the manuscript with great joy and interest! The manuscript is written in a clear and understandable fashion, it is logically sectioned, the methods are explained in appropriate detail and accompanied by custom scripts on a repository, and the analyses are of high scientific quality. There are no major issues that I could identify, but I have some suggestions that may strengthen the paper even further.

1) Relevance for fisheries management

It may be difficult to find the right balance between phrasing the manuscript as a recommendation for fisheries manager or a broadly interesting topic for scientists. I would leave it up to the authors to decide on which aspect they want to focus, but I would recommend to follow this aim throughout the manuscript.

At the moment the title stresses the relevance to fisheries management, but the Introduction is short on any information about current management strategies for Pacific herring. For example, is all Pacific herring along the coast managed as one stock? Or are there distinct management areas? If so, do these areas lump populations that show asynchrony in spawning time together? Is Pacific herring fished throughout the year or are there restrictions during certain times that could protect one but not the other spawning population? Perhaps management areas could also be included in the map on Figure 1a.

Then, for the Discussion I would recommend to come back to the current management strategy outlined in the Introduction and add a brief paragraph on how the here obtained results could be used to improve this strategy. There is an ongoing debate within many conservation research areas on how to bridge the communication gap between scientists and conservation or management practitioners. This lack of communication and lack of mutual understanding was also highlighted in an article about the role of genomics to secure the future of seafood (Bernatchez et al. (2017) *Trends Ecol Evol*, 32, 665-680; <https://doi.org/10.1016/j.tree.2017.06.010>).

So, if fisheries management authorities would approach you and ask for advice on how the findings of this project could be applied to the management of Pacific herring or fish populations in general, what would you recommend? Could for example specific SNPs identified in this study be used to genetically monitor the populations? Or identify and control landings of specific populations? Or could spatial management areas be closed for fishing during certain times to protect distinct populations? Perhaps a brief paragraph on this issue and any actions taken by the authors to bridge the gap between scientists and practitioners could be added in order to support the integration of genomic-informed methods into management and raise awareness of this issue.

2) More on the biology

In the Introduction, I also missed some information about the biology and life history of the Pacific herring populations. Do these populations undergo spawning or foraging migrations or do they remain in their local areas? I guess the eggs and larvae drift with the ocean currents, so for populations to stay differentiated throughout time (some generations?) as shown in this manuscript, they have to return to their birth places or the larvae to be trapped in some sort of local recruitment system. Does natal homing exist in herring? Could you add a short paragraph on this?

3) The issue with DAPCs

Discriminant Analysis of Principal Components (DAPC) is not to be used to test for population structure (see e.g., Miller et al. (2020) *Heredity*, 125, 269–280, <https://doi.org/10.1038/s41437-020-0348-2>). Using DAPC with prior groupings (here: sampling location) will always display cluster according to the sampling locations. That is, because the algorithm tests for differences between the a priori specified groups. Having said this, it is valid to ask whether temporal or spatial distance has the larger influence on population structure in the dataset and use DAPC to answer this question.

I suggest to carefully rephrase all sections concerning the DAPC analyses to explicitly state that the intention is not to identify or visualize population structure, but whether temporal or spatial distance has the larger variance in such.

In addition, I would appreciate if a regular principal component analysis (smartPCA, or "STRUCTURE", or "fineRADstructure", or a similar analysis) could be added, to visualize the general population structure in the data (perhaps remove the highly linked regions for such analyses). Maybe also the F_{ST} distance matrix (Fig. S113) could be used for this, if it is possible to add for example above the diagonal a heat-plot according to high/low F_{ST} and if samples can be clustered according to the strongest pattern.

4) Outlier SNPs in LD region

Among temporally diverged spawning aggregations, several outlier SNPs were identified in a region on chromosome 15. This is a region with high linkage disequilibrium (LD) that may be a chromosomal rearrangement. The strongest outlier are SNPs in a gene that influences the position of retinal photoreceptors, which could be related to light perception and thus regulate photoperiodic reproduction, leading to spawning events at different times of the year.

If this region is in fact a chromosomal rearrangement, recombination is limited in heterozygous individuals and thus any gene within this region could be under selection. What are the other genes in this region? Is there a common function to these genes (i.e., was a gene ontology (GO) analysis done)? Then, the five SNPs in *SYNE2* that are monomorphic in Atlantic herring, are they in the coding region? If so, are these synonymous or non-synonymous substitutions, and do they potentially change the protein structure? If they're not in the coding region or synonymous, they may as well be neutral and do not support parallel selection of this gene.

5) IBD using the shortest distance

For measures of isolation-by-distance (IBD), the shortest distance "as the crow flies" was used, ignoring landmasses and seascapes. Is this a meaningful approach? According to the map, there are many fjords. Possibly the direct line between two locations that are deep in two adjacent fjords (in the fjord part furthest away from the sea) is closer, than a direct line between the mouth and the end of the fjords, although these latter positions would likely experience more exchange than the two locations in different fjords (I hope it is clear what I mean ;)). There would be another method called "least-cost paths", taking landmasses and seascape into account (including bathymetric data) that could be used instead, which would probably be a more realistic measurement for the distance (R package MARMAP (Pante & Simon-Bouhet (2013) PLoS ONE 8, e73051; <https://doi.org/10.1371/journal.pone.0073051>).

Minor comments:

L87-90 How much of the variation in spawning time may be plasticity or epigenetic modifications? Is there some literature on that? Perhaps some information could be added.

L93-95 "Recent studies have demonstrated that chromosomal rearrangements can facilitate local adaptation in species characterized by subtle population structure and/or high levels of gene flow (19, 20)" Such studies are not that recent anymore, e.g. the monkey flower paper by Twyford et al. is from 2015. Perhaps replace "recent studies..." with "studies in various species..." and add an example for insects (e.g., mosquito, *Drosophila*, or *Heliconius*). Furthermore, the paper by Kess et al. focuses on the divergence of Atlantic cod across the species' range and not necessarily on adaptation in local populations with subtle population structure and/or high levels of gene flow. A much better fit would be the studies on Atlantic cod in local fjord systems such as Barth et al. (2019) *Mol. Ecol.* 28, 1394-1411 (<https://doi.org/10.1111/mec.15010>).

L 110-112 This phrasing is a bit misleading. The 6,718 loci are used in the final analyses, the original number of genotyped loci was probably much higher.

L 112-114 I first thought each individual was sequenced again. Perhaps rephrase to: "After filtering for missing data, each individual sample had an average read depth of..."

L138-140 Here I wondered about larval drift, spawning/foraging migrations, and natal homing in Pacific herring. After having added something about this in the introduction, you could refer to it here.

L192 "the same genes", which genes?

L263 Are you sure you used 100% ethanol?

L270 nM should not be italic

L272, 276 Base pair may be abbreviated with bp

L274 For demultiplexing, how many mismatches of the barcodes were allowed? Did you check reads for having the correct cut sites?

L273-304 At all steps, could you add how many loci/SNPs and/or individuals you had before and then after each filtering step or how many were removed, respectively? The default dataset was not filtered for linkage disequilibrium, correct? Could you clarify in which analyses all loci were included and where only unlinked loci were included?

L327 “..incorporating environmental data..”, add “..incorporating environmental data (we used geographic and temporal connectivity)..” and remove this information from L331

L347 only the p in p-value should be italic

L357-359 Using the sampling date? Or the approximate date of spawning? Since some of the samples were collected in pre-spawning condition, the approximate date of spawning may be more appropriate.

L359-362 How much burn-in was excluded?

L376 k in k-means should be italic

L383-385 ...as described before?

L388 remove comma between Pacific herring and sister species

Methods section general (including Supplemental methods): Include version number for each software specified.

References general: There are some references with their title capitalized in title case (e.g., L 469, 509), make sure all titles are small caps (unless names etc.). Also, check that species names are italic (e.g., L469)

Figure 3a (and SI6, 7a): Could the total length of the chromosome and the length of the LD region be added to the Figure?

Figure 3e: Could you indicate which allele frequency differences were identified as significantly different in the Figure?

Supplemental information:

L16 Please include the demultiplexing step and how many mismatches were allowed

L41-47 “in each sampling location” - merging temporal samples from the same location? How many loci were excluded based on HWE filtering? Were loci also excluded if deviating from HWE in only a single population? Were loci excluded showing both heterozygous excess and homozygous deviation? Selection can of course also cause deviations from HWE and if SNPs in even a single population, not checking for homo-/ or heterozygous deviation were removed with a significance threshold of 0.05, important adaptive variation may have been unintentionally removed.

SI Table 1 Please include the month (or period) each population is spawning and the exact date used in the analyses. Also, the abbreviations in the provided VCF file do not fully match the location names given in this table. Please either add the abbreviations to the table or re-name the individuals in the VCF accordingly.

GitHub: The file “Julian_date_metadata.txt” is not provided as SI data nor on GitHub. Perhaps this file and any other data file that might be needed to reproduce the figures using the provided scripts could be added.

Referee: 2

Comments to the Author(s)

The current study aims at studying functional genetic diversity in Atlantic herring, with a focus on reproductive timing. The dataset used here is invaluable, containing over 1100 individuals from 23 spawning locations along the Pacific coast of North America.

Overall the quality of the work is very high and the writing excellent, but I would recommend to reformulate the abstract to better emphasize the implications of the findings for fisheries management. Additionally several full sentences in the beginning of the introduction are exactly the same as in the abstract (l.67-70 and 79-81), so I encourage rewriting either in the abstract or in the introduction.

Another general comment concerns the PCA done with PCAdapt. Here it appears that loci in LD (likely corresponding to inversions here) were kept for the analysis. After reading the manual for

PCAdapt, the authors recommend to remove loci in LD for the analysis. What was the justification to keep them in?

l. 282: why the cutoff of 20,000 loci? How many loci were in the database?

l. 282: Why couldn't the authors use all individuals in cstacks? This would probably bias the SNP identification?

l.284-285: why removing loci that had more than 20% missing data in any one sampling location? Because of potential SNP in restriction site or null allele?

l.294: MAF 0.05 across all samples = over 100 occurrences (50 homozygotes or 100 heterozygotes). So it's unlikely that they are genotyping errors. Maybe they shouldn't be removed when trying to find adaptive loci? See Ahrens et al. 2018.

l.298: What parameters for bowtie2?

l. 342: how many loci were biallelic?

l. 370-371: Are the authors talking specifically about inversions here (based on the wording "alternate orientation")? If that's the case, it should be explicit. There are other forms of rearrangements that do not involve orientation changes (although they do have impacts on recombination).

Author's Response to Decision Letter for (RSPB-2020-2398.R0)

See Appendix A.

Decision letter (RSPB-2020-2398.R1)

28-Jan-2021

Dear Dr Petrou

I am pleased to inform you that your manuscript entitled "Functional genetic diversity in an exploited marine species and its relevance to fisheries management" has been accepted for publication in Proceedings B.

Open Access

Paper charges

Sincerely,

Dr Daniel Costa

Associate Editor:

Comments to Author:

The authors have done an excellent and thorough job addressing the comments of the positive and helpful reviews they received in the original submission. Indeed, the revisions to their genetic cluster analyses and the updates to align the utilities of this approach to fisheries management paint a compelling story of the relevance of functional genomics with these types of questions. The study reads much better as a result. For all of these reasons, I think the study will attract a broad audience and I recommend to accept.

Appendix A

Dear Dr. Costa and Reviewers,

We would like to thank you very much for reading our manuscript and providing us with valuable feedback on how it can be strengthened. We have incorporated your suggested edits whenever it was possible, and we think that the resulting manuscript is greatly improved as a result of your input. Please see our responses to your specific comments below.

Kind regards,

Eleni Petrou and Lorenz Hauser, on behalf of coauthors

Associate Editor Comments

Comments to Author:

This study has now been thoroughly reviewed by two experts in the field. I agree with both Referees that an MS convincingly demonstrating how the genetic structure of Pacific Herring populations is driven by temporal differences in spawning time followed by spatial distance is a significant finding of broad interest to the community, with implications for the role and evolutionary influence of temporal population diversity in natural populations. Thus, the study has the potential to be of broad interest to PRSB.

Nonetheless, the Referees draw attention to certain caveats in the approach and writing that should be addressed to further strengthen the manuscript. First, recent data has drawn attention to some of the limitations of DAPC (Referee 1). The Referee makes an excellent suggestion to clarify that DAPC is being used to assess variance in temporal and spatial distance with this method. I agree additional analyses here (e.g., smartPCA), removing highly linked regions please, would help support the conclusions. In addition, the notion that the results will be applicable to fisheries management falls a little flat and is somewhat unconvincing as presented. Both Referees outline specific scenarios that could be addressed to strengthen this aspect. Overall, I agree that the issue is an important one. I am also very curious about MARMAP to address a more meaningful representation of distance in the IBD analysis (Referee 1). Both Referees make several other important observations and suggestions that should be carefully considered in the revision. For all of these reasons, I am recommending that the MS be revised following these helpful reviews.

- Thank you very much for reviewing our manuscript. Following your and the reviewers' recommendations, we have clarified wording around the DAPC and added a principal components analysis using unlinked markers. We have also added information on herring management to the introduction, and strengthened the link between our results and implications for fisheries management in the discussion. In order to incorporate this additional information to the manuscript and still be within the 10-page limit of the journal, we have moved some technical details on the methods to the electronic

supplementary material. Please see how we incorporated each of the reviewers' suggestions to our manuscript by reading our detailed responses below. We provide updated line numbers that refer to the resubmitted manuscript and electronic supplementary material (without tracked changes) to help orient the reader to these changes.

Referee 1 Comments

Comments to the Author(s)

The manuscript "Functional genetic diversity in an exploited marine species and its relevance to fisheries management" by Eleni Petrou and colleagues uses RAD sequencing of Pacific herring samples to investigate whether spatial and temporal shifts of spawning aggregations have an underlying genetic basis. They find that population structure is majorly driven by temporal differences in spawning time followed by spatial distance, identify genes related to light reception possibly influencing photoperiodic regulation of reproduction, and advice fisheries management authorities to take not only spatial, but also temporal population diversity into account.

I read the manuscript with great joy and interest! The manuscript is written in a clear and understandable fashion, it is logically sectioned, the methods are explained in appropriate detail and accompanied by custom scripts on a repository, and the analyses are of high scientific quality. There are no major issues that I could identify, but I have some suggestions that may strengthen the paper even further.

1) Relevance for fisheries management

It may be difficult to find the right balance between phrasing the manuscript as a recommendation for fisheries manager or a broadly interesting topic for scientists. I would leave it up to the authors to decide on which aspect they want to focus, but I would recommend to follow this aim throughout the manuscript.

At the moment the title stresses the relevance to fisheries management, but the Introduction is short on any information about current management strategies for Pacific herring. For example, is all Pacific herring along the coast managed as one stock? Or are there distinct management areas? If so, do these areas lump populations that show asynchrony in spawning time together? Is Pacific herring fished throughout the year or are there restrictions during certain times that could protect one but not the other spawning population? Perhaps management areas could also be included in the map on Figure 1a.

- Pacific herring are distributed across multiple administrative areas and countries, and so management plans for the species are quite varied and complex. We agree with the reviewer that an overview of herring management would strengthen the manuscript, and so we added a summary of management schemes and user conflicts in the

introduction (please see lines 91-107 in the manuscript, and below). We tried to keep the summary succinct, as we recognize that our manuscript is very close to the word limit:

During the spawning season, Pacific herring support commercially and culturally important fisheries. Commercial fisheries harvest eggs from sexually mature fish prior to spawning, while Indigenous fisheries primarily harvest eggs that have been deposited on vegetation (some adult herring are taken for subsistence). The fisheries are managed spatially, though the geographic extent of spatial units varies by country and administrative area (i.e., state or province). In Washington State, spawning biomass is estimated for individual spawning areas, and a limited bait fishery harvests herring outside the spawning season (22). In British Columbia, spawning biomass is estimated for five major and two minor stocks and used to set annual quotas for the commercial fishery (23). In Alaska, fisheries are managed as regulatory stocks that spawn on specific beaches and coastlines, though regulations vary regionally (24). In general, stocks combine multiple local spawning aggregations, many of which have cultural and economic significance for Indigenous groups (25, 26). This sets up a conflict between resource users when local spawning aggregations decline in abundance or collapse, as spatially constrained groups (e.g. Indigenous fishers) are affected more severely than highly mobile industrial fishing fleets (27). Furthermore, spatial management schemes may not account for temporal population structure, and declines in spawn timing diversity (28) may impact the resilience of this marine resource (29).

Then, for the Discussion I would recommend to come back to the current management strategy outlined in the Introduction and add a brief paragraph on how the here obtained results could be used to improve this strategy. There is an ongoing debate within many conservation research areas on how to bridge the communication gap between scientists and conservation or management practitioners. This lack of communication and lack of mutual understanding was also highlighted in an article about the role of genomics to secure the future of seafood (Bernatchez et al. (2017) *Trends Ecol Evol*, 32, 665-680; <https://doi.org/10.1016/j.tree.2017.06.010>). So, if fisheries management authorities would approach you and ask for advice on how the findings of this project could be applied to the management of Pacific herring or fish populations in general, what would you recommend? Could for example specific SNPs identified in this study be used to genetically monitor the populations? Or identify and control landings of specific populations? Or could spatial management areas be closed for fishing during certain times to protect distinct populations? Perhaps a brief paragraph on this issue and any actions taken by the authors to bridge the gap between scientists and practitioners could be added in order to support the integration of genomic-informed methods into management and raise awareness of this issue.

- We have followed the reviewer's suggestion to interpret how our results could inform herring management schemes in the Discussion. We tried to keep this discussion

succinct because our manuscript is very close to the word limit. We have added the following information to lines 260-266 in the manuscript:

Undetected population structure within management units can lead to ‘cryptic collapses’ (27) that may reduce portfolio effects maintaining population stability at the species level (6). Temporal population structure in spatially managed species may be particularly vulnerable if fisheries targeting, for example, early spawners also accidentally exploit smaller late-spawning populations. Such bycatch could be quantified and considered in assessment by genetic population assignment using some of the highly discriminatory loci (64) identified in this study.

2) More on the biology

In the Introduction, I also missed some information about the biology and life history of the Pacific herring populations. Do these populations undergo spawning or foraging migrations or do they remain in their local areas? I guess the eggs and larvae drift with the ocean currents, so for populations to stay differentiated throughout time (some generations?) as shown in this manuscript, they have to return to their birth places or the larvae to be trapped in some sort of local recruitment system. Does natal homing exist in herring? Could you add a short paragraph on this?

- Following the reviewer’s recommendation, we have added the following information about the biology and life history of Pacific herring to the paper’s introduction (please see lines 83-90 in the manuscript):

On the west coast of North America, Pacific herring migrate to the nearshore environment in winter and spring to spawn on intertidal and subtidal marine vegetation (15). Sexually mature fish gather near the spawning grounds several weeks or months before spawning (15) and quickly disperse after reproducing. The movement and distribution of Pacific herring outside of the spawning season are poorly understood, but there is evidence from contaminants (19) and stable isotopes (20) that some populations migrate offshore to feed while others reside in coastal waters and estuaries. Mark-recapture studies (reviewed in 21) show that Pacific herring display fidelity to relatively broad geographic areas.

3) The issue with DAPCs

Discriminant Analysis of Principal Components (DAPC) is not to be used to test for population structure (see e.g., Miller et al. (2020) *Heredity*, 125, 269–280, <https://doi.org/10.1038/s41437-020-0348-2>). Using DAPC with prior groupings (here: sampling location) will always display cluster according to the sampling locations. That is, because the algorithm tests for differences between the a priori specified groups. Having said this, it is valid to ask whether temporal or

spatial distance has the larger influence on population structure in the dataset and use DAPC to answer this question. I suggest to carefully rephrase all sections concerning the DAPC analyses to explicitly state that the intention is not to identify or visualize population structure, but whether temporal or spatial distance has the larger variance in such. In addition, I would appreciate if a regular principal component analysis (*smartPCA*, or “STRUCTURE”, or “fineRADstructure”, or a similar analysis) could be added, to visualize the general population structure in the data (perhaps remove the highly linked regions for such analyses). Maybe also the *FST* distance matrix (Fig. SI13) could be used for this, if it is possible to add for example above the diagonal a heat-plot according to high/low *FST* and if samples can be clustered according to the strongest pattern.

- We thank the reviewer for bringing these points and the work of Miller et al. (2020) to our attention. We have rephrased the sections of the manuscript pertaining to the DAPC following the suggestions of the reviewer, clarifying that the DAPC was used to assess variance in temporal and spatial distance between sampling locations. We also followed the reviewer’s suggestion and added a principal components analysis using *smartPCA* (please see lines 131-134 in the manuscript). This PCA was conducted using a subset of loci that were pruned for linkage disequilibrium.

4) Outlier SNPs in LD region

Among temporally diverged spawning aggregations, several outlier SNPs were identified in a region on chromosome 15. This is a region with high linkage disequilibrium (LD) that may be a chromosomal rearrangement. The strongest outlier are SNPs in a gene that influences the position of retinal photoreceptors, which could be related to light perception and thus regulate photoperiodic reproduction, leading to spawning events at different times of the year.

If this region is in fact a chromosomal rearrangement, recombination is limited in heterozygous individuals and thus any gene within this region could be under selection. What are the other genes in this region? Is there a common function to these genes (i.e., was a gene ontology (GO) analysis done)? Then, the five SNPs in *SYNE2* that are monomorphic in Atlantic herring, are they in the coding region? If so, are these synonymous or non-synonymous substitutions, and do they potentially change the protein structure? If they’re not in the coding region or synonymous, they may as well be neutral and do not support parallel selection of this gene.

- The referee brings up some very good questions. Our data set included 290 SNPs on chromosome 15, and this chromosome had high linkage disequilibrium that extended over 15.9 Mb (Figure 3B). Despite the presence of this LD block on chromosome 15, we did not observe uniformly high correlations between the allele frequencies of all 290 SNPs and spawning photoperiod. Instead, we only identified 21 SNPs on this chromosome whose allele frequencies were strongly correlated to spawning photoperiod. The strongest outliers were distributed within the gene *SYNE2*. As

recommended by the reviewer, we evaluated whether outlier SNPs were found within coding sequences and whether they represented synonymous or non-synonymous substitutions. All of the SNPs within *SYNE2* that were strongly correlated with spawning photoperiod were found within the coding sequence of this gene. The SNP with the highest correlation to spawning photoperiod (\log_{10} Bayes Factor = 41) was a non-synonymous missense variant (serine to histidine), which is predicted to have a moderate effect on the protein structure. The other outlier SNPs in *SYNE2* are intron variants or splice region variants. Taken together, these data are consistent with selection acting on this particular locus. Of course there may be additional loci under selection on chromosome 15 that we were not able to detect using a RAD sequencing approach. Future studies using whole genome sequencing should be able to fully address this question and precisely identify the precise breakpoints of the putative chromosomal rearrangement. We have added this information to the manuscript (please see lines 211-216 in the manuscript), to clarify our reasoning on these points.

5) IBD using the shortest distance

For measures of isolation-by-distance (IBD), the shortest distance “as the crow flies” was used, ignoring landmasses and seascapes. Is this a meaningful approach? According to the map, there are many fjords. Possibly the direct line between two locations that are deep in two adjacent fjords (in the fjord part furthest away from the sea) is closer, than a direct line between the mouth and the end of the fjords, although these latter positions would likely experience more exchange than the two locations in different fjords (I hope it is clear what I mean ;)). There would be another method called “least-cost paths”, taking landmasses and seascape into account (including bathymetric data) that could be used instead, which would probably be a more realistic measurement for the distance (R package MARMAP (Pante & Simon-Bouhet (2013) PLoS ONE 8, e73051; <https://doi.org/10.1371/journal.pone.0073051>).

- We liked the referee’s suggestion of using a least cost path analysis to estimate the waterway distance between spawning locations. As the referee suggested, we used the MARMAP package in R to download bathymetric data from NOAA using the highest resolution possible (resolution = 1). Unfortunately, the geography of the coastline in our study area is very intricate and bathymetric data layers are inaccurate for the nearshore environment where herring were sampled. For example, NOAA’s bathymetric data do not have enough resolution to identify that there is a major marine waterway (Johnstone Strait, indicated by black arrow in figure below) between Vancouver Island and the BC Coast. As a result of this error, spatial distances between herring in Puget Sound and the northern BC coast are overestimated, and two of the inlet sampling locations (Bute Inlet and Knight Inlet) appear to be totally landlocked. Furthermore, the bathymetric data erroneously indicate that the many of our sampling locations in the

Puget Sound are also on land. Given the constraints and errors of these bathymetric data, we think that the linear distance between sampling locations along a linear coastline is a better approximation of the geographic distance separating sampling locations.

Minor comments:

L87-90 How much of the variation in spawning time may be plasticity or epigenetic modifications? Is there some literature on that? Perhaps some information could be added.

- To address these questions, we conducted a literature search on Google Scholar and Web of Science. We did not find any published literature that has evaluated the role of epigenetic modifications on the reproductive timing of herring. However, we did find a very interesting recent study by Berg et al. (2020) that quantified whether Atlantic herring from spring and autumn-spawning stocks switch their spawn timing using genetics and otolith microchemistry. This study demonstrated that a small percentage (8%) of herring switch between spawning seasons, but the majority display high fidelity to spawning season. We have added a citation to this article in the introduction (please see line 109 in the manuscript).

L93-95 “Recent studies have demonstrated that chromosomal rearrangements can facilitate local

adaptation in species characterized by subtle population structure and/or high levels of gene flow (19, 20)” Such studies are not that recent anymore, e.g. the monkey flower paper by Twyford et al. is from 2015. Perhaps replace “recent studies...” with “studies in various species...” and add an example for insects (e.g., mosquito, *Drosophila*, or *Heliconius*). Furthermore, the paper by Kess et al. focuses on the divergence of Atlantic cod across the species' range and not necessarily on adaptation in local populations with subtle population structure and/or high levels of gene flow. A much better fit would be the studies on Atlantic cod in local fjord systems such as Barth et al. (2019) *Mol. Ecol.* 28, 1394-1411 (<https://doi.org/10.1111/mec.15010>).

- We thank the referee for bringing this literature to our attention. We have changed the wording following the referee’s suggestions, have added a reference to inversions and local adaptation in mosquitoes (Ayala et al. 2013) and have also cited the paper by Barth et al. (2019) (please see lines 110-111 in the manuscript).

L 110-112 This phrasing is a bit misleading. The 6,718 loci are used in the final analyses, the original number of genotyped loci was probably much higher.

- We have added detailed information on locus and SNP discovery in the electronic supplemental materials (please see lines 182-206 in the electronic supplementary materials) and we have added a reference to that information in the manuscript to avoid misleading the reader with our phrasing.

L 112-114 I first thought each individual was sequenced again. Perhaps rephrase to: “After filtering for missing data, each individual sample had an average read depth of...”

- We have changed this wording to improve clarity as recommended by the referee.

L138-140 Here I wondered about larval drift, spawning/foraging migrations, and natal homing in Pacific herring. After having added something about this in the introduction, you could refer to it here.

- We have added information on how the interannual consistency of allele frequencies within a particular spawning population supports the hypothesis that Pacific herring home to their natal spawning site (please see lines 157-160 in the manuscript).

L192 “the same genes”, which genes?

- To improve clarity, we have rephrased this sentence as: *In Atlantic herring, a specific subset of genes related to the photoperiodic regulation of reproduction are highly associated with spawn timing in populations from both sides of the Atlantic Ocean.*

L263 Are you sure you used 100% ethanol?

- Yes, our lab has a practice of placing tissue samples in 100% ethanol (200 proof & molecular-grade from Sigma Aldrich). We do this to avoid low ethanol concentration resulting from the high water content in fish tissue.

L270 nM should not be italic

- We have fixed this following the referee's suggestion.

L272, 276 Base pair may be abbreviated with bp.

- We have fixed this following the referee's suggestion.

L274 For demultiplexing, how many mismatches of the barcodes were allowed? Did you check reads for having the correct cut sites?

- Sequences were checked for having the correct cut site and we allowed for one mismatch between barcodes. We have added these details to the electronic supplementary material (please see lines 21-24).

L273-304 At all steps, could you add how many loci/SNPs and/or individuals you had before and then after each filtering step or how many were removed, respectively? The default dataset was not filtered for linkage disequilibrium, correct? Could you clarify in which analyses all loci were included and where only unlinked loci were included?

- We have added detailed information about the number of loci, SNPs, and individuals retained after each filtering step in the electronic supplemental materials (please see lines 182-206 in the electronic supplementary materials). We have also clarified that the *smartPCA* analyses used the LD-pruned set of loci, while all other analyses used the full set of loci (line 204 in the supplemental materials).

L327 “..incorporating environmental data..”, add “..incorporating environmental data (we used geographic and temporal connectivity)..” and remove this information from L331

- We have edited these sentences following the reviewer's suggestion.

L347 only the p in p-value should be italic

- We have fixed this following the referee's suggestion.

L357-359 Using the sampling date? Or the approximate date of spawning? Since some of the samples were collected in pre-spawning condition, the approximate date of spawning may be more appropriate.

- We used the sampling date for this analysis and we have clarified this point in the manuscript (please see line 91-93 in the electronic supplementary material). For samples collected after 2014, this date is also the spawning date because we sampled during active spawning events. We don't have a way to accurately estimate the spawning date of samples collected prior to 2004 (these were sexually mature herring collected from "pre-spawning holding areas"), so we used the sampling date as a proxy for the spawning date.

L359-362 How much burn-in was excluded?

- In the *bayenv2* program, the user does not specify a burn-in period.

L376 k in k-means should be italic

- We have fixed this following the reviewer's suggestion.

L383-385 ...as described before?

- We have added additional details on how we tested for deviations from HWE to the manuscript (please see lines 360-363).

L388 remove comma between Pacific herring and sister species

- We have fixed this following the referee's suggestion.

Methods section general (including Supplemental methods): Include version number for each software specified.

- We have added software version numbers to the manuscript and to the electronic supplementary materials.

References general: There are some references with their title capitalized in title case (e.g., L 469, 509), make sure all titles are small caps (unless names etc.). Also, check that species names are italic (e.g., L469)

- We have corrected these mistakes following the reviewer's recommendations.

Figure 3a (and SI6, 7a): Could the total length of the chromosome and the length of the LD region be added to the Figure?

- Following the reviewer's recommendation, we have annotated the length of the LD region in the figures and added information on the total length of the chromosome to the relevant figure legends.

Figure 3e: Could you indicate which allele frequency differences were identified as significantly different in the Figure?

- Our initial analysis used a chi-squared test to reject the null hypothesis that the frequencies of the inversion genotypes were the same across all spawning groups. To address the reviewer's question and identify statistically significant allele frequency differences across different hierarchical levels, we replaced the chi-squared test with an analysis of molecular variance (AMOVA). We have added the AMOVA results as SI Table 3 in the electronic supplementary materials and also refer to them in the manuscript (please see lines 192-200).

Supplemental information:

L16 Please include the demultiplexing step and how many mismatches were allowed

- Following the referee's recommendation, we have added this information to the electronic supplementary material (please see lines 21-24).

L41-47 "in each sampling location" - merging temporal samples from the same location? How many loci were excluded based on HWE filtering? Were loci also excluded if deviating from HWE in only a single population? Were loci excluded showing both heterozygous excess and homozygous deviation? Selection can of course also cause deviations from HWE and if SNPs in even a single population, not checking for homo- or heterozygous deviation were removed with a significance threshold of 0.05, important adaptive variation may have been unintentionally removed.

- We have clarified that loci were tested for deviations from Hardy-Weinberg equilibrium (HWE) in each sampling location and sampling year (we did not merge temporal samples from the same location). To create a conservatively filtered data set, we excluded loci if they had a q -value less than 0.05, even if this deviation occurred in a single population. We identified 620 loci that were out of HWE using these criteria. We re-did specific analyses of population structure (DAPC, isolation by distance and time, and F_{ST} estimation) using this filtered data set as a quality control check, to verify that the patterns of population structure in our data were not primarily driven by loci that were out of HWE. For analyses of adaptive variation (*bayenv2*), we used the full data set

that included all 6,718 loci because we reasoned that some loci might be out of HWE due to very strong selection. We have added this information to the electronic supplementary material (please see lines 47-55).

SI Table 1 Please include the month (or period) each population is spawning and the exact date used in the analyses. Also, the abbreviations in the provided VCF file do not fully match the location names given in this table. Please either add the abbreviations to the table or re-name the individuals in the VCF accordingly.

- We thank the referee for bringing this to our attention. We have clarified the caption of SI Table 1, explaining that the column “Sampling_Date” shows the date of sampling an active spawning aggregation of herring (after 2014) or the date of sampling sexually mature herring in pre-spawning aggregations (before 2014). This “Sampling_Date” is also the exact date used in all analyses. We have also fixed the naming discrepancy between SI Table 1 and the VCF file.

GitHub: The file “Julian_date_metadata.txt” is not provided as SI data nor on GitHub. Perhaps this file and any other data file that might be needed to reproduce the figures using the provided scripts could be added.

- We thank the reviewer for bringing this to our attention. The file “Julian_date_metadata.txt” has now been added to GitHub.

Referee: 2

Comments to the Author(s)

The current study aims at studying functional genetic diversity in Atlantic herring, with a focus on reproductive timing. The dataset used here is invaluable, containing over 1100 individuals from 23 spawning locations along the Pacific coast of North America.

Overall the quality of the work is very high and the writing excellent, but I would recommend to reformulate the abstract to better emphasize the implications of the findings for fisheries management. Additionally several full sentences in the beginning of the introduction are exactly the same as in the abstract (l.67-70 and 79-81), so I encourage rewriting either in the abstract or in the introduction.

- We thank the reviewer for their feedback. Following their suggestions, we have edited the abstract to emphasize the implications of our study for the delineation of fishery management units (please see lines 41-42). We have also edited the opening sentences of the abstract to avoid repetition with the first sentences in the introduction.

Another general comment concerns the PCA done with PCAdapt. Here it appears that loci in LD

(likely corresponding to inversions here) were kept for the analysis. After reading the manual for PCAdapt, the authors recommend to remove loci in LD for the analysis. What was the justification to keep them in?

- Our goal in using *pcadapt* was to look for SNPs that were outliers in regards to overall population differentiation in our samples. We have clarified this point in lines 71-76 in the electronic supplementary materials). Specifically, we wanted to look for the interaction of loci identified as being under selection by *bayenv* and more general outliers identified by *pcadapt*. This was motivated in part by studies that have demonstrated that chromosomal rearrangements can facilitate local adaptation in species characterized by subtle population structure and/or high levels of gene flow (for example, please see references 31-33 in the manuscript). For this reason, we decided to retain all loci in the *pcadapt* analysis. We do agree though that if we had exclusively used *pcadapt* to infer local adaptation (and did not compare the results with another method like *bayenv*), then it would be better to prune loci for LD to avoid a high number of false positives.

l. 282: why the cutoff of 20,000 loci? How many loci were in the database?

- We apologize that our wording was not sufficiently clear here. After sequences were assembled into putative alleles (“stacks”) in *pstacks*, we found that the mean number of stacks per sample was 27,052 (sd = 3,671). However, a small number of samples (N = 52) contained far fewer stacks (mean = 12,056; sd = 6,964). We filtered out these low-quality samples by removing individuals in the 1% quantile of the distribution (i.e. containing fewer than 20,000 stacks). We have added this information to the electronic supplementary materials (please see lines 183-187).

l. 282: Why couldn't the authors use all individuals in *cstacks*? This would probably bias the SNP identification?

- The reviewer brings up a good question. Because of the large number of samples in our study (N = 1,344), it was computationally prohibitive to build a catalog using all individuals (in other words, *cstacks* would crash without outputting any data because our computer would run out of RAM). As Rochette & Catchen (2017) have shown that the number of shared loci in the catalog quickly plateaus as more samples are included in *cstacks*, we decided to use the 10 most deeply sequenced individuals from each collection site ($N_{\text{Stacks}} = 260$) to build the catalog. We have clarified this point in the electronic supplementary materials (please see lines 188-194).

Rochette, N., Catchen, J. Deriving genotypes from RAD-seq short-read data using *Stacks*. *Nat Protoc* 12, 2640–2659 (2017). <https://doi.org/10.1038/nprot.2017.123>

1.284-285: why removing loci that had more than 20% missing data in any one sampling location? Because of potential SNP in restriction site or null allele?

- Yes, that is exactly correct. We were concerned about the impact of null alleles and large amounts of missing data on subsequent population genetic analyses (PCA, etc.). We used this filter to try and remove those potential null alleles from our data. We have clarified this point in the electronic supplementary material (please see lines 33-34).

1.294: MAF 0.05 across all samples = over 100 occurrences (50 homozygotes or 100 heterozygotes). So it's unlikely that they are genotyping errors. Maybe they shouldn't be removed when trying to find adaptive loci? See Ahrens et al. 2018.

- Filtering for minor allele frequency can remove sequencing errors that can bias tests for selection (Roesti et al. 2012) but it can also remove rare alleles that are less informative for estimating population genetic parameters such as F_{ST} (Hendricks et al. 2018). While Ahrens et al. 2018 bring up the excellent point that rare variants have been identified as causative agents in human gene-trait association studies, this is in the context of slightly deleterious SNPs that are subject to weak purifying selection and contribute to human disease. Also Ahrens et al. (2018) acknowledge that “a low MAF may contribute to inconstant population structure estimates (De la Cruz & Raska, 2014; Mathieson & McVean, 2012). This can be overcome by removing minor alleles below the 0.05 threshold when estimating population structure.” Since our study was primarily focused on population genetic questions, we opted to follow this more conservative filtering approach, recognizing that this may come at the cost of failing to identify some true rare alleles that may be under selection.

Hendricks, S, Anderson, EC, Antao, T, et al. Recent advances in conservation and population genomics data analysis. *Evol Appl.* 2018; 11: 1197– 1211. <https://doi.org/10.1111/eva.12659>

Roesti, M., Salzburger, W. & Berner, D. Uninformative polymorphisms bias genome scans for signatures of selection. *BMC Evol Biol* 12, 94 (2012). <https://doi.org/10.1186/1471-2148-12-94>

1.298: What parameters for bowtie2?

- Following the reviewer's recommendation, we have added this information to the manuscript (please see line 310).

1. 342: how many loci were biallelic?

- All of the SNPs in the final filtered data set are biallelic. We have clarified this information in the electronic supplementary material (please see line 198).

l. 370-371: Are the authors talking specifically about inversions here (based on the wording “alternate orientation”)? If that’s the case, it should be explicit. There are other forms of rearrangements that do not involve orientation changes (although they do have impacts on recombination).

- We have followed the reviewer’s recommendation and edited these lines to clarify that we are discussing chromosomal inversions (please see electronic supplementary material, line 104-106).